# Microbial Biofilms: Applications, Clinical Consequences, and Alternative Therapies

**DOI:** 10.3390/microorganisms11081934

**Published:** 2023-07-29

**Authors:** Asghar Ali, Andaleeb Zahra, Mohan Kamthan, Fohad Mabood Husain, Thamer Albalawi, Mohammad Zubair, Roba Alatawy, Mohammad Abid, Md Salik Noorani

**Affiliations:** 1Clinical Biochemistry Lab, D/O Biochemistry, School of Chemical and Lifesciences, Jamia Hamdard, New Delhi 110062, India; mohan.kamthan@jamiahamdard.ac.in; 2Department of Botany, School of Chemical and Lifesciences, Jamia Hamdard, New Delhi 110062, India; andaleebzahra2000@gmail.com; 3Department of Food Science and Nutrition, College of Food and Agriculture Sciences, King Saud University, Riyadh 11451, Saudi Arabia; fhusain@ksu.edu.sa; 4Department of Biology, College of Science and Humanities in Al-Kharj, Prince Sattam Bin Abdulaziz University, Al-Kharj 11942, Saudi Arabia; t.albalawi@psau.edu.sa; 5Department of Medical Microbiology, Faculty of Medicine, University of Tabuk, Tabuk 71491, Saudi Arabia; mohammad_zubair@yahoo.co.in (M.Z.); dr.roba2010@hotmail.com (R.A.); 6Department of Biosciences, Jamia Millia Islamia, New Delhi 110025, India; mabid@jmi.ac.in

**Keywords:** quorum sensing, quorum quenching, phage therapy, anti-virulence compounds, antimicrobial peptide, plant extracts, anti-biofilm, biofouling, bioremediation, clinical settings

## Abstract

Biofilms are complex communities of microorganisms that grow on surfaces and are embedded in a matrix of extracellular polymeric substances. These are prevalent in various natural and man-made environments, ranging from industrial settings to medical devices, where they can have both positive and negative impacts. This review explores the diverse applications of microbial biofilms, their clinical consequences, and alternative therapies targeting these resilient structures. We have discussed beneficial applications of microbial biofilms, including their role in wastewater treatment, bioremediation, food industries, agriculture, and biotechnology. Additionally, we have highlighted the mechanisms of biofilm formation and clinical consequences of biofilms in the context of human health. We have also focused on the association of biofilms with antibiotic resistance, chronic infections, and medical device-related infections. To overcome these challenges, alternative therapeutic strategies are explored. The review examines the potential of various antimicrobial agents, such as antimicrobial peptides, quorum-sensing inhibitors, phytoextracts, and nanoparticles, in targeting biofilms. Furthermore, we highlight the future directions for research in this area and the potential of phytotherapy for the prevention and treatment of biofilm-related infections in clinical settings.

## 1. Introduction

Biofilms have gained popularity due to their extreme resistance to removal and treatment. Their resistant nature was brought to light by Characklis in 1973 [1]. They involve an immobile single- or sometimes multispecies colony that includes bacteria, fungi, diatoms, and protozoa that attach to living and non-living surfaces, i.e., are sessile [1,2]. William J. Costerton, in the year 1978, used the term biofilm for the first time [3]. Biofilms have gained importance in recent times because of their role in medical and pharmaceutical sectors as they cause several diseases in human and animal hosts, as well as contamination of medical implants and other equipment [4,5]. Biofilms are expensive in terms of both money and human life. It is estimated that out of all hospital infections, 65% are caused due to biofilms [6]. Apart from health, biofilms are detrimental to food-based industries, including, but not limited to, the dairy processing, brewing, seafood, meat, and poultry industries [7]. They also have a negative impact on the drinking and healthcare water distribution systems as they cause bio-corrosion of pipes, degradation of water quality, and hence, outbreaks of water-borne diseases [8,9]. The ability of microorganisms to form biofilms is known to have a beneficial effect in the agriculture sector, where biofilms offer plant protection and bioremediation of soil [8]. Biofilms are also known to aid in corrosion inhibition and wastewater treatment. As a result, they have a broad application spectrum in the field of biotechnology [8]. The NBIC Annual report (National Biofilms Innovation Centre’s annual Report 2022) provides data about the economic impact of the biofilms, which is represented in the form of a pie chart in Figure 1. An estimated USD 4 tn of value is associated with biofilms. It is seen that the highest cost is from corrosion, which accounts for about 69%, and then health, which has about a 10% contribution [10].

Current estimates suggest that about 40–80% of bacterial cells on Earth can form biofilms [8]. The aggregation of bacteria precedes the formation of biofilms [11]. The aggregation of bacteria can either be surface-associated, as in the biofilms developed on medical implants, which is responsible for causing diseases such as prosthetic valve endocarditis, biofilms on orthopedic and dental implants, peritoneal dialysis catheters, and urinary catheters, or it could be non-surface-associated, as seen in biofilms developed in chronic infections such as cystic fibrosis (CF), and bacterial aggregates discovered from the ocean, freshwater, and water treatment systems [11,12,13]. If the collection involves genetically similar bacteria, it is called auto-aggregation, and if the bacteria are different, it is termed co-aggregation [11]. The biofilm development is determined by several conditions of the environment, such as pH, temperature, salinity, ionic strength, rate of flow of the medium, i.e., hydrodynamics, and nutrient availability [14]. It is also affected by the properties of the surface on which it is formed. Its formation is promoted by the hydrophobic nature of the carrier, increased roughness, and non-polarity of the surface [14,15]. Bacterial cell-surface characteristics such as the hydrophobicity, free energy, overall charge, topography, and geometrical features of the bacterial cell surface also affect its attachment and development [14]. The production of chemicals such as c-AMP (cyclic adenosine monophosphate), c-di-GMP (bis-(3′-5′)-cyclic-dimeric guanosine monophosphate), quorum sensing signal molecules, and EPS (extracellular polymeric substances) also determine cell attachment and growth [15,16]. The formation of biofilms provides a survival advantage as compared to the planktonic state by safeguarding the organism from changes in its environmental conditions. They are protected from the environment and are resistant to all kinds of stressors, such as desiccation, shear stress, toxins, and grazing by microfauna due to the presence of EPS, which constitutes 90% of the mass of dry biofilms and is the principal constituent of biofilms [11,17]. EPS is primarily composed of high-molecular-weight heteropolysaccharides arranged in linear or branched chains, along with extracellular DNA and glycoconjugates. This is collectively referred to as the matrixome, which forms a slippery surface by combining with water, and also forms a complex 3D architecture [3,17,18,19]. EPS varies with the bacterial composition and the stage of development. Biofilms are also known to retain certain special substances, such as particles of clay, blood, corroded fragments, and certain mineral crystals. This varies based on where the biofilms originated [15].

These colonies also promote immense cell-to-cell interactions as QS (quorum sensing) [1]. QS systems regulate bacterial behaviour in the population through signalling mediated by autoinducer molecules. The AI-2 and AI-3 (autoinducers 2 and 3) quorum sensing is used in both Gram-positive and Gram-negative species. In Gram-negative bacteria, QS signalling is mediated by AHLs (N-acyl homoserine lactones), also known as AI-1 (autoinducer 1). It is the best-studied signalling mechanism and is described in Figure 2a [20]. The AIP (autoinducing peptides) molecules are secreted by Gram-positive bacteria. AIP are small peptides that are processed post-translationally and are impermeable to the cell membrane [4]. The two-component pathway in Gram-positive bacteria is illustrated in detail in Figure 2b [20]. Autoinducers are released by bacterial cells in response to population density and accumulate in their environment. When the concentration of these QS molecules attains a threshold value, the AHL molecules, which are transcription factors, attach to the LuxR receptors and regulate the expression of certain genes that are responsible for synthesising virulence factors, such as exotoxins, enzymes, biofilms, and others [21]. The communication between microbes can alter gene expression in response to the concentration of QS molecules, and thereby regulate community behaviour [15,22]. The regulation of gene expression can, in turn, affect the virulence, pathogenicity, and ultimately, the survival of bacteria [23].

To facilitate the research, a repository of all QSSM (quorum sensing signalling molecules), called SigMol, has been created. It contains information related to different families of signalling molecules, such as AHLs, HAQ (4-hydroxy-2-alkylquinolones), DPK (diketopiperazines), DSF (diffusible signal factors), AI-2, AI-3, QSPs (quorum sensing peptides), and many others [24].

It is now known that biofilm development occurs in a step-by-step pattern, known as the attached growth process, as follows: -reversible binding to the surface due to the presence of cations, adhesion that is irreversible, followed by the formation of a microcolony and the maturation of the colony, which disperses as free planktonic cells [15,16,19]. According to a survey, biofilm research has seen an upward trend in the last decade. The search with the keyword “biofilm” in the title yielded over 72,500 publications from 1975 to 2022 on PubMed, of which over 55,600 articles have been published in the last decade, from 2012 to 2022. On browsing the search engine VentureRadar with the keyword “biofilm”, a list of 227 companies was displayed. This indicates the growing commercial interest and also highlights the need for bench-side-to-bedside research in the field of biofilms and the development of anti-biofilm agents [25]. This growth in biofilm research has been triggered by the advancement and development of several techniques of molecular biology, such as microscopic, spectroscopic, bioinformatics, sequencing, and several other methods, which are summarised in Figure 3 [16,26,27,28,29]. Novel methods such as SAW (surface acoustic waves) and bioimpedance-based sensing are also used [30].

The goal of the present review is to provide a comprehensive understanding of different sectors where biofilm formation plays an active role, either in a positive or negative way, and additionally gain insights into different plant derivatives that can act as potential anti-biofilm agents.

## 2. Biofilm and Its Development

Biofilms are a predominant form of microbial growth. Over time, several perspectives have been put forward to explain the nature of biofilms. The most simplistic and commonly accepted model is to consider biofilms as aggregation of individual cells which are formed in a definite five-step process, as shown in Figure 4. The three major processes are attachment, colony formation, and dispersion [1].

The prevailing model of biofilm development, consisting of five sequential steps, has been established using in vitro experimental conditions and is primarily focused on the growth of *Pseudomonas aeruginosa*. This model has also been extrapolated and widely accepted for the study of biofilm development in *Staphylococcus aureus* [11]. It delineates the lifecycle of biofilm-forming organisms into two distinct stages: the motile stage and the sessile stage [31]. The process begins by attachment of free-living microbes to the biotic or abiotic surface, reversibly, via non-specific interactions such as Vander Waals forces and ionic interactions. The attachment becomes irreversible once the adhesins and adhesive proteins are made and production of cyclic c-di-GMP begins. The levels of intracellular c-di-GMP govern the nature of bacterial existence in a planktonic or biofilm state. These cells then form micro-colonies that undergo maturation by EPS formation. The matured biofilms disperse some of the cells that serve as inoculum to spread the biofilm. This process is also called seeding dispersal [32]. Dispersion mainly occurs due to nutrient deficiency, mechanical stress, formation of flagella, degradation of EPS, or the formation of toxins [33].

Recent studies have pointed out disparities in the model and its limited applicability in vivo and in medicinal and industrial settings [11]. Thus, Sauer et al. have proposed an alternative, expanded, and open model of biofilm development that actually displays the bacteria as having the potential to switch forms in response to the factors such as the nature of the substrate, the colonising bacteria, and the conditions in the micro-environment [11]. Bacteria under in vivo conditions can be present in a planktonic or colonial form, as illustrated in Figure 5 [11].

Another approach is to consider biofilms as having characteristics similar to multi-cellular organisms. This is considered since cells in biofilm exhibit coordinated responses, and individual cells behave as differentiated ones in displaying the division of labour. In addition to this, different parts of biofilms have specific roles, such as the functioning of organ systems. They can also control the behaviour and maintain a stable internal environment via QS signalling. The QS system is responsible for interactions between cells, which can be positive as well as negative [31].

Penesyan et al. provided an in-depth insight into the above-mentioned approaches and proposed a novel vision that attempts to fill in the gaps and provide a holistic understanding of the unique nature of biofilms [31]. They stated that biofilms provide a protected environment for cells to accumulate mutations and incorporate genetic and phenotypic changes that enhance their fitness in response to environmental stress [31].

## 3. Biofilms and CRISPR-Cas System

CRISPR (Clustered Regularly Interspaced Short Palindromic Repeats) are a family of DNA repeats widely distributed in prokaryotes. A CRISPR loci consists of a short sequence, 21–48 bp, repeated several times—about 250 times. The *cas* gene (CRISPR-associated gene) is located adjacent to the CRISPR loci [34]. The CRISPR-Cas system serves as a prokaryotic defence mechanism against plasmids and phages attacking the cell [35,36]. CRISPR/Cas adopts at least two basic mechanisms: (1) by acquiring proto-spacers from foreign DNA at the leader end of the CRISPR locus (adaptation stage), and (2) by targeting either invasive DNA or RNA (interference stage) [36]. There is growing evidence that biofilm formation is regulated by the CRISPR-Cas systems that exist in bacteria. The studies suggest an interrelation between the genes that are responsible for biofilm formation and the CRISPR-Cas system. It has been implicated in the regulation of bacterial physiology, virulence, pathogenicity, and the formation of EPS [37]. The CRISPR system can be used as a safe and targeted approach to treat microbial infections. It requires specific cleavage of the Cas9 complex component signalling system, which is a regulator of bacterial virulence [37]. The expressions of CRISPR-associated genes and proteins are seen in various Gram-positive and Gram-negative bacteria associated with humans. By controlling gene expressions of several virulence genes, several biofilm-related diseases can be cured. Researchers have been able to target this system for developing anti-biofilm strategies. Zuberi et al. introduced the concept of CRISPRi (CRISPR inhibition). This process can produce many levels of gene knockdown [38]. It targets the LuxS gene, which regulates quorum signalling. The LuxS gene that synthesises AI-2, a molecule which initiates biofilm formation, was made to hybridise with complementary “single-guide RNA” (sgRNA), and hence, its expression was silenced [39].

## 4. Economic Importance of Biofilms

### 4.1. Biofilm in Environment

Food-processing industries face major risks due to the formation of microbial biofilms in large equipment as well as in food spoilage [7]. QS-signalling-mediated biofilms severely impact the food industries as bacteria are known to cause food spoilage and are responsible for several food-borne diseases, as majority of the biofilm-formers are human pathogens [4]. It is estimated that 60% of all food-borne outbreaks are due to biofilms [8]. Biofilms can be formed on substrates such as wood, glass, rubber, and steel, and hence, cause damage to machineries [40]. This creates technical challenges for the food-processing industries [8].

Most of these biofilms are composed of several mixed species. Hence, it becomes extremely important to understand the diverse nature of biofilms impacting the fresh produce, dairy, meat and fish processing, seafood, fermentation, and brewing industries [41]. The control of biofilms is necessary in ensuring the safety of food in its various stages, such as production, storage, and distribution. This points to the need for more research in food microbiology [41]. Common pathogens of the food industry include *Escherichia coli, Bacillus cereus, Campylobacter jejuni, Yersinia enterocolitica, Listeria monocytogens,* and certain species of *Staphylococcus* and *Salmonella* [7]. Fresh fish products can become contaminated with biofilms of *Aeromonas hydrophila*, *L. monocytogenes*, *S. enterica*, or *Vibrio* spp., that lead to serious health problems [42]. Biofilms can play a positive role in the fermentation process. The biofilm-mediated microbial fermentation is efficient in biochemical production because of the unique property of cell immobilisation, resistance to toxic compounds, and maintenance of long-term cell activity [43]. Microbial groups present in fermented food items (such as beers, wines, distillates, meats, fishes, cheeses, and breads) remain inside the biofilm and are engulfed in EPS, which provides favourable growth conditions to the inhabiting species [44].

In the current global scenario, agriculture stands as the predominant economic sector of India. Following the green revolution, India witnessed a significant surge in the production of food grains, primarily attributed to the widespread adoption of high-yielding varieties (HYV) seeds, accompanied by the escalated utilisation of chemical fertilisers, pesticides, and herbicides [22]. However, this rise in agricultural productivity has led to a decline in soil quality and a negative impact on human health. Hence, the farming industry needs to be developed in a sustainable and environmentally friendly way in order to feed the world’s expanding population [45]. An innovative method for sustainable farming is through the application of biofilm fertilisers [46]. Biofilms produced by AIMs (agriculturally important microbes) can revolutionise the concept of sustainable agriculture [47]. The AIMs include soil microbiomes that participate in the biogeochemical cycles [26]. An example includes PGPR (plant growth-promoting rhizobacteria). It is a specialised class of bacteria which successfully colonises the roots of the host plant and aids in plant development and productivity by providing plant growth hormones, ammonia, enzymes, and other secondary metabolites. PGPRs are often utilised bio-fertilisers [19]. PGPR is an umbrella term that includes many bacteria, such as *Rhizobium*, *Gluconacetobacter*, *Pseudomonas, Azotobacter*, *Azospirillum,* and several others [19].PGPR also helps in bioremediation and biocontrol agents.

Natural biofilm fertilisers are made of a single species. Recently, there has been a rise in curiosity about multispecies biofilms, particularly those involving bacteria with fungi, as commonly seen in mycorrhiza [47]. Compared to monoculture biofilms, the development of mixed-species biofilms has additional benefits, including the production of unique polysaccharides and improved soil ecology [47]. Bacterial biofilms can interact with plants in intricate ways that are mutualistic, commensal, or pathogenic [45]. Bacteria are known to colonise almost all parts of the plant and cause bacterial diseases. Many of these disease-causing bacteria are known to form biofilms on the plant surface as well as in the rhizosphere [19]. Soil health plays an important role in regulating the growth of plants. The ability of microbes to colonise roots and form biofilms can be exploited to improve the supply of nutrients, inhibit the occurrences of diseases, and protect plants. There is evidence that microbes transferred from healthy soil to degraded soil can improve soil health [19]. The formation of bacterial biofilms in soil is influenced by various factors. Among these, edaphic factors such as soil pH and nutrient levels, as well as environmental factors such as temperature and oxygen levels, play crucial roles. Additionally, enzymes and antimicrobial chemicals also significantly contribute to the regulation of bacterial biofilm formation in soil [47].

Biofilms have assumed importance in the degradation of organic pollutants, mainly because they provide eco-friendly, low-cost, and green technology [15]. Microbial bioremediation is possible due to the presence of several active functional groups that have been found to be present on the surface of biofilms. Functional groups are known to promote diffusion through the biofilm, which aids in bioremediation [15]. Several microbes belonging to species of *Pseudomonas, Arthrobacter, Alcanivorax, Cyclocasticus, Bacillus,* and *Rhodococcus* have the ability to clear out hydrocarbons in marine settings that are present because of petroleum-based industries. In combination with surfactants, multispecies biofilms can easily clear the crude oil spills [15].

The bioremediation of wastewater is perhaps the most important use of biofilms. According to a report, India generated an estimated amount of 1.12 × 10^11^ litres of wastewater per day in the year 2020–2021, with double the amounts coming from urban settings as compared to rural areas [48]. With these increasing levels of water, the demand for clean water is on the rise. Wastewater that contains microbial biofilms poses a serious health hazard but in contrast these microbial biofilms appear to have the potential to treat the wastewater [49]. The microbial biofilms are being employed in the current technologies used for wastewater treatment as they are known to help in the removal of contaminants. Wastewater, particularly released from agriculture and industries, is often rich in nitrogen [50]. Excessive nitrogen is known to be responsible for severe environmental consequences, such as algal blooms and eutrophication of water bodies, and subsequently poses a risk to aquatic life, as well as being a human health hazard and human life [51]. In N-rich wastewater treatment, the ability of bacteria to remove contaminants is used by the process of activated sludge, and recently through biofilm bioreactors. These processes can be promoted by adding inoculants of bacteria, a process known as bioaugmentation. These bacteria can remove nitrogen through nitrification, denitrification, partial nitrification–anaerobic ammonium oxidation, and partial denitrification–anaerobic ammonium oxidation [50,52]. As with nitrogen, a high quantity of phosphate may also choke water bodies by promoting unnecessary growth in water bodies. Phosphate-accumulating organisms participating in biofilm formation may help in the removal of this growth. For this, microalgal biofilms composed of *Chlorella vulgaris*, *Scenedesmus vacuolatus*, and *Scenedesmus obliquus* have been found to be efficient in this process [53].

Moreover, biofilms are industrially utilised for the production of biogas, which is a source of sustainable energy, through anaerobic digestion [54]. Technologically advanced wastewater treatment facilities based on biofilms are increasingly being built across the world as they provide several advantages over the use of the free-planktonic forms of bacteria used previously. Several bacteria capable of degrading different contaminants can persist together in mixed-species biofilms [29]. Hence, it offers the concurrent removal of multiple pollutants from wastewater at once. In addition, the loss of important bacteria and biomass can be avoided [29]. Biofilm-based water treatment plants are also cost-effective and energy-efficient [53]. Algal biofilms can grow on wastewater, making it free from heavy metals and pollutants. They can then be harvested and used for the production of biofuels [17]. In the future, biofilm-treated wastewater might be helpful for crop irrigation as a source of clean water in natural water bodies during drought-like situations [53]. The upcycling of wastewater is necessary to achieve the SDGs (Sustainable Development Goals) [48]. A recent study reported the effect of biofilms on the RNA of SARS-CoV-2. The biofilms in sewer water can alter the stability of the viral RNA and promote its degradation, and in contrast, may sometimes aid in the prolonged prevalence of the SARS-CoV-2 RNA in wastewater [55]. *Acenatobacter radioresistens*, *Bacillus subtilis*, and surfactant have together been shown to degrade crude oil spills [15].

Lastly, biofouling caused by bacterial biofilms of the surface results in the loss of several billion dollars globally. As the name states, biofouling means polluting of the surface due to biological agents. Biofouling starts with bacterial biofilms, that pave the way and make surfaces suitable for colonisation by other protozoa, algae, and diatoms. This is especially a problem for man-made surfaces submerged in salty seawater conditions, such as military and civilian marine vessels [56]. The marine conditions provide a favourable environment for the microbial biofilm to develop and enhance corrosion of the metallic surfaces, referred to as MIC (microbially influenced corrosion) [57]. It is also referred to as biological corrosion and is mainly due to the activity of sulphur-reducing bacteria, sulphur-oxidising bacteria, and manganese- and iron-oxidising and reducing bacteria. Bio-corrosion is responsible for 20% of all corrosion in aqueous environments [14]. It results in the formation of a slimy layer on ship hulls and pipelines, an increase in hydraulic resistance, and affects the fuel consumption and energy costs involved [58]. Anti-fouling substances can be developed by coating surfaces with plant-derived anti-adhesive substances. The anti-fouling feature conferred by adhesin inhibition can be investigated for safety and toxicity and utilised to develop safe medical implants [59].

### 4.2. Biofilms in Health

Human health is closely linked to the presence of bacteria in the human gut. Human health is affected by pathogenic bacteria that are responsible for causing numerous diseases, such as dental plaques, cystic fibrosis, infective endocarditis, urinary tract infections, ventilator-associated pneumonia, chronic wounds, chronic sinusitis, otitis media, and periodontitis [54]. The microbes associated with important infections are listed in Table 1. These are called BAIs (biofilm-associated infections) [30]. The formation of biofilms, both single-species and multispecies, presents a major challenge in disease treatment. Antimicrobial resistance is reportedly responsible for 700,000 deaths annually, and by 2050, that number is expected to be 10 million according to the United Nations Interagency Coordinating Group on AMR [60]. Abolition of established bacterial biofilms is difficult since they become resilient to even antiseptics and disinfectants as well as the immune system and drugs due to the presence of a thick matrix [6]. The pathogenicity associated with biofilms is a consequence of their resistance to antibiotics, and the resistance has a genetic basis [61]. MDR (multidrug resistance) genes have been identified in several bacterial species that confer the ability to hinder the mechanism of drug action [61]. Many variables contribute to the development of resistance. Common mechanisms of resistance found in bacteria include the alteration of target proteins by mutation, drug inactivation by enzymes, acquisition of genes from other species that express less susceptible proteins, and avoiding drugs to access the target sites [62]. Sakarikou et al. listed the following reasons for drug resistance in biofilms: antibiotics are unable to penetrate the EPS and reach the planktonic cells therein, differences in gene expression in response to stress, unevenness of surfaces, inactivation of antimicrobial chemicals within EPS, altered environmental conditions within the biofilms, and increased activity of efflux pumps which remove the antibiotics [23]. The major facilitator superfamily (MF), the resistance-nodulation-division family (RND), the small multidrug-resistance family (SMR), the ATP-binding cassette family (ABC), and the multidrug and toxic compound extrusion family (MATE) are five different classes of bacterial efflux pumps that have been identified [63]. The formation of dormant cells and persister cells in response to stress and the slow growth of cells are major reasons for the development of resistance, and can lead to chronic infections [33,61,64]. The inefficiency of antibiotics in disease treatment due to biofilm resistance can hamper the global health system and have severe consequences on public health at large. This highlights the need for better and alternative approaches to combat biofilms via development of anti-biofilm compounds.

The biofilms associated with living organisms are not always bad. Certain microbes are essential for the healthy functioning of the human body. Probiotic biofilms are known to have bacteria that are beneficial for human gut health. They help in the growth of tissues and benefit the immune system. These include *Lactobacillus*, *Bacillus laterosporus,* and *Pediococcus acidilactici* [61].

#### 4.2.1. Device-Related Biofilm Infection

##### Urinary Tract Infection (UTI)

The most prevalent bacterial illnesses are UTIs, which are also the most frequently diagnosed urological diseases. Compared to men, women are thought to be more affected by UTIs. UTIs are commonly caused because of the insertion of catheters during hospital stays. They can also develop as a result of germs travelling from the urethra to the bladder. Some individuals have a genetic tendency to develop UTIs. Bacterial strains that are Gram-negative are more frequently responsible for this infection [65]. UPEC (uropathogenic *E. coli*) strains have been found to be most commonly responsible for uncomplicated cases, particularly possible as they exist in the intestinal area but may migrate to the urinary tract and turn pathogenic. They show type 1 fimbriae, P. fimbriae, flagellum, capsular lipopolysaccharide, and proteins associated with the outer side of the cell membrane as bacterial cell surface virulence factors. Haemolysin and siderophores are secreted virulence factors [66].

##### Nosocomial Infections

Nosocomial infections develop in healthcare settings. The bacteria of the ESKAPE group are the pathogens that are responsible for major infections developing in care units, prone to become multidrug-resistant (MDR). The ESKAPE group includes *Enterococcus faecium*, *Staphylococcus aureus*, *Klebsiella pneumoniae*, *Acinetobacter baumannii*, *Pseudomonas aeruginosa*, and *Enterobacter* species [67]. Majority of healthcare-associated infections (HAI), about 60–70%, are known to be of medical device-, biomaterial-, and implant-related origins [68]. These are known as BRDIs (biofilm-related device infection). These include biofilms due to implants: contact lenses, prosthetic valves, urinary catheters, peritoneal dialysis catheters, intravascular catheters, cerebrospinal fluid shunts, prosthetic joints, pacemakers, endotracheal tube, voice prostheses, mechanical heart valves, breast implants, and biliary stents [6,8]. Infection of implants is difficult to be cleared by the host defence because the presence of a foreign object creates a local region of depressed immune activity [69]. More than 500,000 different kinds of medical devices are currently on the global market, according to previous estimates from Medtech Europe and FDA162 [69]. There are approximately 10 million procedures for dental implants [69]. Additionally, frequently touched objects including doorknobs, switches, and railings are known to be the cause of infections. Recently, water has also been highlighted as a source of HAI outbreaks [70].

##### Breast Implant Infection (BII)

In patients undergoing breast surgical procedures, the colonisation of the surgical site by bacteria and the development of biofilms is common and important [71]. Worldwide, about 5–10 million women currently have artificial breast implants [69]. The biofilm formation leads to the development of capsular conjuncture in about 5.2–30% of patients with breast implant surgeries [72]. It has also been implicated in the development of BI-ALCL (breast implant-associated anaplastic large cell lymphoma) [71]. BIA-ALCL is a rare type of lymphoma associated with textured breast implants. Studies suggest that biofilm formation on the surface of breast implants may play a role in developing this condition. This biofilm provides a favourable environment for bacterial growth and can trigger chronic inflammation and immune responses. It is believed that the chronic inflammation associated with biofilm infection may contribute to the development of BIA-ALCL [73]. In implant culture, *Staphylococci* has been frequently isolated. *S. aureus* and *S. epidermidis* and other anaerobes colonise breast implants and are responsible for infections and implant loss [74]. *Pseudomonas aeruginosa* has also been found to be responsible for the biofilm formation. These microorganisms can create biofilms on the implant surface, which increases their resistance to both human immunological defences and medicines. Investigations are ongoing to determine the precise pathways by which biofilm infection contributes to the emergence of BIA-ALCL. According to a theory, biofilm-induced chronic inflammation may culminate in immunological dysregulation, genetic changes, and ultimately, lymphoma formation. Important factors to consider include biofilm infection prevention and control in breast implants. The likelihood of biofilm formation can be reduced by employing methods such as appropriate surgical techniques, antibiotic prophylaxis, and routine monitoring for infection symptoms. Implant removal may be required in cases when biofilm infection is suspected or confirmed to address the infection and lower the risk of lymphoma.

##### Catheter-Related Bloodstream Infection

In healthcare settings, catheter-related bloodstream infections (CRBSIs) present a serious problem since they raise morbidity, mortality, and medical expense rates. They are considered as the most common nosocomial infections, and the risk of infection increases with longer hospital stays [75]. About 400,000 cases of CRBSIs are reported annually [76]. The development of biofilms on the surface of intravascular catheters, such as central venous catheters (CVCs) and arterial catheters (Acs), is one of the major causes of CRBSIs [77]. Microorganisms can flourish and survive better in the biofilm matrix, increasing their resistance to antimicrobial treatments. They function as a physical barrier to protect the pathogens from the human immune system and to stop antimicrobial medications from penetrating. Due to the biofilms’ toughness, it is often necessary to remove the infected catheter to completely eradicate CRBSIs [78]. The material of the catheter, the method of insertion, the length of the catheterisation, and the existence of underlying medical problems are only a few of the variables that affect biofilm formation on catheters [79]. *Staphylococcus epidermidis*, *Staphylococcus aureus*, *Enterococcus* species, and *Candida* species (*C. albicans* and *C. parapsilosis*) are typical microbial species linked to CRBSIs [75,80]. The effects of biofilm development on catheters are severe. They can result in systemic infections such as endocarditis, septicaemia, and persistent bloodstream infections. Further complicating the clinical course of the infection, biofilms can also act as a reservoir for the spread of germs to other body regions. In order to lower the prevalence of CRBSIs, it is essential to prevent biofilm growth on catheters. To reduce the production of biofilms, methods such as stringent aseptic insertion techniques, good hand hygiene, catheter site care, and antimicrobial catheter coatings have been investigated [80]. Coating the catheters with anti-infective substances has been seen to decrease the microbial contamination [81].

##### Periprosthetic Joint Infection (PJI)

A side effect of joint replacement surgery is periprosthetic joint infection (PJI) [82]. Developing biofilms on the surface of prosthetic joints is crucial to the pathogenesis of PJI. Complex microbial communities called biofilms attach to artificial surfaces encased in a protective matrix, which makes them extremely resistant to drugs and the immune system. *Staphylococcus aureus*, *Staphylococcus epidermidis*, and *Enterococcus* species are typical microorganisms linked to PJI [83]. Biofilms on prosthetic joints cause persistent and recurrent infections, which have a negative impact on patient outcomes and necessitate revision surgery. Chronic inflammation, implant loosening, and tissue damage brought on by biofilms can result in pain and functional impairment [84]. The fact that biofilms can withstand antimicrobial treatments makes controlling PJI extremely difficult. There are numerous prevention and therapy options for PJI caused by biofilms. They entail strictly adhering to sterile surgical procedures, choosing the right antimicrobial prophylaxis, and providing the best perioperative care [85]. Additionally, cutting-edge strategies are being investigated to battle biofilm formation and improve treatment outcomes, such as the use of antimicrobial coatings on prosthetic surfaces and the creation of biofilm-disrupting agents [86].

##### Contact Lens Infections

People who wear contact lenses have good reason to be concerned about infections caused by their lenses. The development of biofilms on contact lenses is a typical occurrence in these illnesses, which can aid in the growth and duration of infections. Complex populations of microorganisms known as biofilms are coated in a protective matrix and stick to surfaces, including contact lenses. Microorganisms are shielded by this biofilm matrix, making them more resistant to antimicrobial treatments. Biofilms may be formed in vivo on the posterior surface of the contact lens or the lens storage cases [87]. The nature of the lens material, how the lenses are cared for, and the environment around the eyes are only a few of the variables that might affect the development of biofilms on contact lenses [88,89]. Bacterial species commonly associated with contact lens-related infections include *Staphylococcus aureus*, *Pseudomonas aeruginosa*, and *Serratia marcescens*. Additionally, fungal species such as *Candida albicans, Aspergillus*, and *Fusarium* species can form biofilms on contact lenses [90,91]. Forming biofilms on contact lenses can lead to adverse effects, including inflammation, corneal ulcers, and impaired vision. Ocular infections such as microbial keratitis (MK), infiltrative keratitis (IK), contact lens-induced peripheral ulcer (CLPU), and contact lens-induced acute red eye (CLARE) may occur [92]. The IK, CLPU, and CLARE are together known as erythrogenic or conjunctival inflammation [92]. To prevent contact lens-related biofilm formation, various strategies can be employed. These include maintaining proper lens hygiene, practicing regular disinfection of lenses, and utilising antimicrobial agents. Use of multipurpose contact lens solutions, antimicrobial and biocidal coatings on surfaces, and surface modifications can help in reducing biofilm formation [93,94]. Advances in material science have facilitated the development of new contact lens materials with improved surface properties, aiming to reduce biofilm formation and its associated risks. *Calendula officinalis* and *Buddleja salviifolia* extracts have shown promising results against biofilms on soft contact lenses [88].

##### Ventilator-Associated Pneumonia (VAP)

Patients using mechanical ventilation are susceptible to a serious infection known as ventilator-associated pneumonia (VAP). A significant role in the emergence and persistence of VAP is biofilm growth on the surfaces of endotracheal tubes (ETT) and ventilators [95]. The onset of VAP is seen after 48 h of mechanical breathing [96]. Bacteria can develop in a supportive environment thanks to the biofilm matrix, which enables them to attach to surfaces and create intricate structures. *Staphylococcus aureus*, *Klebsiella pneumoniae*, *Pseudomonas aeruginosa*, and *Acinetobacter baumanii* are common microorganisms linked to biofilm-related VAP [97]. These biofilms can potentially cause persistent infections, higher death rates, and lengthier hospital admissions [98]. Biofilms on the ventilator and endotracheal tube surfaces provide a source for ongoing bacterial colonisation and encourage bacterial aspiration into the lower respiratory tract. This then sets off an inflammatory reaction and impairs lung function. The capacity of biofilms to withstand antimicrobial treatments makes managing VAP even more challenging. Strategies for biofilm-related VAP prevention and control are essential for lowering its prevalence. These tactics include strictly adhering to infection control procedures, regularly checking endotracheal tubes for the development of biofilms, and maintaining good oral hygiene habits. In addition, surface modification through various antimicrobial coatings on ETT and biofilm disruption technologies are being researched as potential therapies [99,100].

#### 4.2.2. Tissue-Related Biofilm Infections

##### Dental Biofilms

Microorganisms can develop in a peculiar environment in the human mouth. The teeth, dental implants, and other tissues provide diverse ecological niches for the oral microbes to create biofilms. The oral cavity supports a balance of natural flora that exist as biofilms. These biofilms house an estimated number of 700 distinct species of bacteria, 100 species of fungi along with several viruses that form a complex web of interaction [101]. Inadequate maintenance of oral hygiene and negligence of oral health can lower the pH of the oral cavity, and this disrupts the ratio of the oral microbiome, i.e., dysbiosis leads to infections [101]. This situation turns the commonly non-pathogenic *Candida* species into pathogenic, which leads to fungal infections. In severe circumstances, these biofilms can cause diseases such as oropharyngeal candidiasis (a fungal infection), bacterial infections including dental caries, with more than 3.5 billion cases worldwide, and periodontal diseases, and diseases related to oral implants [102]. *Bacillus gaemokensis* is the major biofilm-former in dental carries [102]. *Streptococcus mutans* and *Filifactor alocis* are the bacteria that generally cause biofilm development in periodontitis [103]. Peri-implantitis and peri-implant mucositis are common infections associated with dental implants [104].

##### Cystic Fibrosis (CF)

More than 160,000 people around the world are estimated to be living with cystic fibrosis. The virulence of *Staphylococcus aureus* and *Pseudomonas aeruginosa* has been known to develop biofilm in the lungs [105]. This genetic condition affects the cystic fibrosis transmembrane conductance regulator protein, and is characterised by a cycle of infection and inflammation which has a severe negative impact on the patient’s ability to breathe. The community-associated methicillin-resistant (MRSA) strains are of particular danger as they produce a wide range of virulence factors in the form of toxic proteins and immune-evasive factors, and they are associated with mortality and morbidity [106].

##### Infective Endocarditis (IE)

Infective endocarditis is a life-threatening cardiovascular infection occurring on the endocardium, artificial valves, artificial implanted devices, and on the inner surface of the heart [107]. Cases of infective endocarditis have been on the rise in the past 3 decades, from 478,000 cases in 1990 to 1,090,530 cases in 2019, with about 25% mortality [108]. About 80% of IE cases are attributed to *Streptococcus*, *Staphylococccus*, and *Enterococcus* [108]. The biofilm formation in case of infective endocarditis begins by the attachment of microbes to the surface of a prosthetic valve, the damaged endocardium of the heart, or the valve sub-endothelium, with the help of polysaccharides and fibronectin [109]. Treatment of IE with the help of antimicrobials has become challenging due to the protection offered to the underlying pathogens by the biofilm [69]. Surgical removal of the biofilms remains the final option for curing IE [109]. Gilbey et al. have reported a case where phage therapy has been used on patients suffering from *Staphylococcal* sepsis and prosthetic wall endocarditis [110].

##### Chronic Wound Infections (CWI)

Chronic wounds present a major burden on healthcare [111]. Wounds may show delayed healing, mainly due to underlying causes such as diabetes, obesity, hypertension, malignancy, old age, or peripheral vascular disorder [112]. They remain in their non-healing state mainly because they become arrested in any one of the four stages of the wound-healing process (haemostasis, inflammation, tissue proliferation, and tissue remodelling) [113]. Studies have reported that 60–90% of chronic wounds are associated with biofilms of several different pathogenic bacteria, such as *Proteus* spp., β-haemolytic *Streptococci*, Coagulase-negative *Staphylococci*, bacteria of the ESKAPE group, and fungi (*Candida* spp.) [114]. The biofilms form in response to favourable conditions, such as moist, nutrient-rich environments, presence of necrotic debris, a lack of oxygen tension, and a lack of immune response [115]. The bacterial aggregates are associated with the granulation tissue and are dispersed in the cells, such as fibroblasts and keratinocytes, as well as elastin, collagen, and fibronectin, which make up the extracellular matrix [111].

**Table 1 microorganisms-11-01934-t001:** Common biofilm-forming pathogens.

Disease	Pathogens	Reference
Urinary tract infections	*E. coli*, *Klebsiella pneumoniae*, *Proteus mirabilis*, *Pseudomonas aeruginosa*, *Staphylococcus* (*S.aureus*, *S. saprophyticus*, *S. epidermidis*), *Enterococci*, *Streptococci agalactiae*, *Corynebacterium urealyticum*, *Candida*	[65]
Oral health problems (dental plaques, dental carries, and periodontitis)	*Neisseria*, *Granulicatella*, *Streptococcus*, *Actinomyces*, *Veillonella*	[116]
Nosocomial infections (healthcare-acquired infections)	*Staphylococcus epidermidis*, *Candida albicans,* *Staphylococcus aureus*, *P. aeruginosa*, *Klebsiella pneumonia*, *Enterococcus faecalis*, *Proteus mirabilis*	[67]
Sexually transmitted diseases (STDs)	*Neisseria gonorrhoeae*	[117]
Cystic fibrosis	*Pseudomonas aeruginosa* (infects adults), *Staphylococcus aureus* (infects children)	[118]
Infective endocarditis	*Streptococci, Staphylococci, Enterococci*	[109]

## 5. Methods of Combating Biofilms

According to the WHO Global Antimicrobial Resistance and Use Surveillance System (GLASS 2022) report, about 4.95 million deaths were associated with AMR (antimicrobial resistance) in 2019, and AMR may involve up to 3.8% of the GDP (GLASS, 2022). Multidrug resistance is increasing and presents itself as a major global threat to the life of all plants and animals, including humans [119]. MDR naturally develops in pathogenic agents in order to enhance their fitness and aid in survival, and it is increasing at a tremendous pace [119]. Besides the exploitation and misuse of antimicrobials, which is the primary driver, the World Health Organisation (WHO) identifies a lack of cleanliness and sanitation, no proper access to clean water, and a lack of disease control and prevention as contributors to the increasing levels of antimicrobial resistance and tolerance [120]. The enormous spread of resistance traits has resulted in a loss of effectiveness of antibiotics and other antimicrobials [119]. Plasmids are found to be the culprits in the spread of resistance traits against last-resort antibiotics as they aid in horizontal gene transfer for evolution [121]. This indicates that alternative strategies need to be developed to control the spread of such organisms. Since biofilm formation is associated with high resistance of cells to antimicrobials, compounds that have the ability to inhibit biofilms can help us treat such infections. Biofilm-forming bacteria are known to develop 1000-times higher resistance to antibiotics than planktonic-state bacteria [20]. Bacteria inside a biofilm can withstand up to 1000 times the minimum inhibitory concentration (MIC) of antibiotics [33]. This is because biofilms allow limited diffusion of antibiotics owing to the low membrane permeability and the lower number of porins on the outer surface [33]. Several different approaches have been proposed to combat biofilm development. These include the use of antiseptics, disinfectants, antibiotics, bacteriophages, enzymes, essential oils, surface modifications, and QS inhibitors [41]. In order to aid researchers in developing anti-biofilm agents, a database known as aBiofilm has been created. It has structural, biological, and chemical details of all the anti-biofilm methods that have been reported. It can prove to be a very helpful resource to researchers trying to develop and find newer approaches to prevent biofilm formation [122]. Some of the anti-biofilm methods are explained further in this paper.

### 5.1. Phytoextracts

The huge diversity of plants on Earth is a reservoir that is yet to be completely explored. Several plant extracts have been tested in vitro for their potential to cure biofilm-related infections. Studies have shown that plant extracts rich in secondary metabolites and bioactive compounds offer possible treatments for biofilms. There are several plants, the extracts from which are potentially anti-biofilm compounds. Antimicrobial and antifungal compounds from plant extracts have been tested for their effectiveness against bacteria and fungi, both of which can form biofilms [123,124]. In oral health, oral fungal infections are a common occurrence, especially due to *Candida* species. Different species of *Candida* are involved in the occurrence of periodontal infections. These fungal biofilms were tested for the effects caused by treatment with plant extracts. The essential oils extracted from bulbs of *Allium sativum* L. and leaves of *Cinnamomum zeylanica* Blume. and *Cimbapogon citratus* (DC. Stapf.) have demonstrated the potential to fight against infections caused by the opportunistic pathogen *Candida,* or majorly *C. albicans,* which causes Candidiasis and Candidemia when it reaches the bloodstream [101]. The bioactive compounds in these plant extracts have antifungal and anti-biofilm activity. It was seen that the number of *C. albicans, C. dubliniensis,* and *C. tropicalis* was significantly reduced by the combination of glycolic extracts of quercetin from *Rosa centifolia* (white rose) and curcumin of *Curcuma longa* (turmeric)*. C. albicans* and *C. krusei* biofilms were eliminated by the glycolic extract of p-coumaric acid in *Rosmarinus officinalis* (rosemary) and gallotannins of *Punica granatum* (pomegranate) [125]. The chloroform extract of *Piper betle* is reported to be efficacious in both preventing biofilm formation and eliminating existing biofilms of *Bacillus gaemokensis* [102]. Rosemary essential oil was found to be capable of lowering the biofilm produced by *E. coli* by 86.36% in patients of varying ages. Alcoholic phytoextracts have also been reported to have proven antibacterial action [126].

Studies have shown promising results, indicating that certain phytoextracts possess antibacterial properties and can effectively inhibit the growth of bacteria. Mehmood et al. reported a significant zone of inhibition (ranging from 10 to 24 mm) for plant extracts (aqueous, ethanol, and methanol) against *S. aureus* and *E. coli* standard isolates. These extracts demonstrated notable MIC values ranging from 78 to 625 µg/mL [127]. Dahiya and Purkayastha compared the efficacy of phytoextracts to standard antibiotics and observed a zone of inhibition of 25 mm for Neem (*Azadirachta indica*) ethanolic extract and of 21.9 mm for *Bryophyllum* (Kalanchoe) methanolic extract against *Staphylococcus aureus* ATCC25923, while vancomycin resulted in a zone of inhibition of 21.6 mm. Additionally, ethanolic extract of Tulsi (*Ocimum sanctum*) and *Aloe vera* showed effectiveness comparable to vancomycin against multidrug-resistant *Staphylococcus aureus* MRSA [128]. Essential oil of *Ocimum gratissimum* exhibited low MIC values of 6 and 0.75 µg/mL against *E. coli* and *Staphylococcus aureus,* respectively [129]. Phytoextracts contain bioactive compounds such as alkaloids, flavonoids, terpenoids, and phenolic compounds, which exhibit antimicrobial activity [127,130]. It is important to note that while some phytoextracts may exhibit antibacterial activity, their efficacy and spectrum of action may vary compared to standard antibiotic treatments [131,132].

The antibiofilm ability of *Capsicum baccatum* var. *pendulum*, a member of the Solanaceae family, was investigated against *Pseudomonas aeruginosa* and *Staphylococcus epidermis*. The residual aqueous extract from seeds (RaQS extract) was found to inhibit the biofilm formation, primarily due to its ability to prevent adhesion without affecting the planktonic bacteria [59]. The phytoextracts and volatile essential oils are known to hamper biofilm production using several different mechanisms, such as damaging the membrane and the membrane proteins forming leaky channels and inhibiting ATP production, interrupting the signals, and thereby the communication between cells [133,134]. Plants produce a variety of secondary metabolites that are useful against a variety of human pathogenic bacteria. *Glycyrrhiza* species, a member of the Fabaceae family, contains glycyrrhizin, a metabolically inactive compound. Under in vivo conditions, it is converted into the metabolically active, 18β–glycyrrhetinic acid (GRA). The antibacterial and anti-biofilm activity of the active form against *Neisseria gonorrhoeae* has been demonstrated by Zhou et al. Its ability to inhibit biofilm formation has also been tested for *Streptococcus mutans, S. sobrinus,* and *P. aeruginosa* [117]. *Panax quincuefolius* root extract was found to be effective in lowering the virulence expression and mitigating bacterial swarming/swimming motility, resulting in reduced biofilm formation by *P. aeruginosa*. The extract inhibited bacterial growth and the minimum inhibitory concentration of the root extract was found to be 12,500–25,000 µg/mL [135].

At a 1% *v/v* concentration, the essential oils (EOs) extracted from *Piper nigrum* and *Mentha suaveolens* were found to diminish *Staphylococcus aureus* biofilm development by 40%, mainly by decomposing EPS and dismantling the surface, as demonstrated by scanning electron microscopy (SEM), but not by inhibiting bacterial growth. Eugenol and β-caryophellene were found to be responsible for the anti-biofilm effect [118]. Table 2 presents details of the efficacy of the tested plant extracts against well-known biofilm-forming bacteria. It reveals that different extracts of the same plant can inhibit biofilms at different concentrations (MIC).

### 5.2. Nanoparticles against Biofilms

The biofilm-related microbial infections can be targeted using nanotechnology. It is a non-conventional and effective approach that needs to be tested by clinical trials [141]. It provides a method to treat a broad range of infections by acting as a mode of delivery to particular locations in optimal quantity and enhancing the antimicrobial potential of the drugs. The nanoparticles (NPs) serve as conveyers of biofilm EPS disrupters [142]. The small size, high sensitivity, and large surface area-to-volume ratio are some features that make nanoparticles suitable for penetrating and destroying biofilms [143]. They also protect the drugs from enzymatic reactions [33]. Nanoparticles have not been used in a full-fledged manner due to their toxicity to normal cells and tissues, lack of stability under in vivo conditions, less absorption, and insolubility, which leads to precipitation and aggregate formation [143]. Figure 6 shows the mechanism of action of nanoparticles.

Ongoing studies in the field are focused on addressing these issues [143]. Metals (such as silver, zinc, copper) and metal oxides (zinc oxide, iron oxide) are commonly used nanoparticles (Table 3). When these nanoparticles come in contact with the surface of biofilms, their interactions with the functional groups and surface charges turn them into toxic ions, and they can degrade the EPS and the bacterial cells [143]. During interactions of nanoparticles with the biofilm, the EPS presents as the initial point of contact [144]. This process of interaction is a three-step process: transport of NPs to the target site, their deposition, and the attachment to the surface and migration within the biofilms [145]. This process is regulated by several factors, such as the physiochemical characteristics of the biofilm surface, the charge density, distribution, and heterogeneity [144]. The surface charge, hydrophobicity, functional groups, and size of NPs also influence the NP–biofilm interaction [142,145]. The nanoparticles act via various mechanisms, such as enzyme disruption, DNA damage, cell membrane breakage, peptidoglycan and protein denaturation, plasmid nicking, and oxidative damage [146]. Another nano-based technology is magnetic hyperthermia, that uses the heating potential of metal oxides to destroy the biofilms [33].

### 5.3. Antimicrobial Peptide (AMP)

AMPs contain 5–50 amino acid chains and are made out L-amino acids and form secondary structures, such as alpha helix and beta-pleated sheets [157]. The AMPs are mostly positively charged chains and have a molecular mass of 2–10 kDa. The AMPs are naturally present in animals, plants, and humans as the first line of body defence, i.e., innate immunity against pathogens, and display a broad range of antibacterial activity [158]. There are different approaches that can be utilised to battle against the adhesive nature of biofilms by using AMP. Yasir et al. summarised that AMP can act through mechanisms such as disrupting and degrading the membrane potential of microbial cells constituting the biofilms, interfering with the bacterial QS signalling system, destroying the EPS, and decreasing the levels of (p)ppGpp or alarmone, that are responsible for regulating the bacterial stress response [68,159].

AMPs have several advantages. They have the potential to be utilised against both multidrug-resistant (MDR) and extensively drug-resistant (XDR) bacteria [158]. The AMPs have both a hydrophobic and hydrophilic nature that imparts them amphipathic properties. This makes the AMPs efficient to bind to the lipopolysaccharide of the cell membrane via the Van der Waal’s interaction, penetrate the biofilm, and destroy the bacterial cells. In this way, AMPs are effective in removing infections [158]. An estimated number of 3000+ antimicrobial peptides have been isolated from the enormous diversity of life forms that exist on Earth. Plant-produced AMPs are showing promising results against human pathogens, and therefore, helping in the treatment of bacteria-borne infections. The different parts of plants produce several kinds of peptides. As of January 2023, the antimicrobial peptide database contains a record of 2569 AMPs obtained from different classes of organisms, of which 371 AMPs have been isolated from plant sources [160].

Microbes are observed to be inefficient in developing resistance to AMPs due to their mechanism of action. AMPs can be combined with antibiotics and bioactive molecules to combat biofilms. They can also be combined with matrix-inhibiting compounds such as sulfhydryl compound and iron chelators, or matrix-disaggregating compounds such as NO and chelating agents [60]. Novel strategies such as the use of nanoparticles to deliver AMPs have shown promising results in curing mammalian infections [68,161]. Nevertheless, certain drawbacks have also been reported. Mass production of AMPs is not economically feasible, there is a lack of pharmacokinetic and pharmacodynamic studies on the effects of AMPs, and there is a susceptibility to bacterial protease, which are some limitations presented in the use of AMPs for treating biofilm-based infections [157].

### 5.4. Anti-Virulence Compounds from Plants

Virulence factors are known to be responsible not just for the survival of bacterial cells but also for the infection process. Adherence factors such as fimbriae or pili, swarming motility, siderophores, and many other virulence factors, allow the formation of biofilms [162]. There are several anti-virulence compounds that only inhibit the viral and pathogenic substances without affecting the pathogenic bacteria. This makes the use of this agent mostly free from the development of microbial resistance, with minimal effect on the host microbiome. It can be used synergistically with other methods for immunocompromised patients, where the immune system does not possess enough capability to remove the pathogenic bacteria at all. It is known that the QS system is responsible for the expression of virulence factors; hence, inhibiting the quorum sensing can prevent the expression of several genes that are involved in biofilm production [119,163]. To interrupt the biofilm formation, quorum quenching enzymes or quorum sensing inhibitors can be utilised. These are chemicals that eukaryotes and prokaryotes create in order to disrupt the signalling system [164]. They can also be chemically synthesised [4]. They function by decreasing the activity of AHL synthase, inactivating the AHL molecules, or activating them and creating antagonists that can compete with signal molecules [164]. Different steps of biofilm formation can be targeted using various approaches, such as decreasing the QS signalling, and preventing the synthesis of EPS by blocking e-DNA, protein, and polysaccharide formation [119]. Several plant-derived substances, such as sulphur-containing compounds, monoterpenes and terpenoids, phenylpropanoids, benzoic acid derivatives, diarylheptanoids, coumarin, flavonoids, and tannins, have proven QS-inhibiting properties. They act by directly or indirectly inhibiting the QS system [165].

A recent study has reported a novel anti-virulence product, DAA (dehydroabietic acid), which is a diterpene. It is a natural substance obtained from extracts of Pine with antiviral, antibacterial, and antifungal properties. The artificially synthesised DAA derivatives, particularly those with an amino-alcohol moiety, have established anti-biofilm action. The derivatives are active against blight-causing *Xanthomonas oryzae* pv. *oryzae*, as they inhibit the expression of several virulence factors [166]. Ethanolic extracts of *Persicaria maculosa* and *Bistorta officinalis* have anti-virulence capabilities against *Pseudomonas aeruginosa.* They inhibit biofilm formation by affecting the swimming motility of pathogenic bacteria and are not toxic to the host [167]. This suggests the possible use of plants as the primary source of extracting anti-virulence and anti-biofilm compounds, although more research is needed. Most of the published results are for in vivo experiments, and proper results for in vitro models have not yet been achieved [163].

### 5.5. Phage Therapy

The ability of phages to infect Gram-positive as well as Gram-negative bacteria, their abundance in nature, and limited number of phage receptors on eukaryotic organisms, make them attractive for development as anti-biofilm agents [119,168]. Phages can easily penetrate cells through the water channels, which are actually meant to allow the diffusion of nutrients [169]. Bacteriophages also produce exopolysaccharide-degrading enzymes such as polysaccharide depolymerase, that are present at tail ends and lysins, and they can destroy biofilms [33,170,171]. Other enzymes and their functions are mentioned in Figure 7 [169]. The effect of phages on the host organism and its immune system, the method of administration, the required dose, the development of resistance, the formulation of a phage mixture to combat multispecies biofilms, and the lack of proper clinical trials are a few major challenges in the development [119]. It is speculated that the combination of antibiotics with phage therapy can deliver positive results [172]. Successful results have been obtained by administration of *Staphylococcus aureus*-specific bacteriophage, Sb-1, in curing diabetic foot ulcers (DFU) that were due to *S. aureus* infection [170,173]. In 2006, the FDA (Food and Drug Administration) approved the use of phages against *Listeria* spp., a food-borne pathogen, for use in packaged meat and cheese [170].

## 6. Future Directions

There are several future directions for research in biofilm-related infections. The improvements can be made the in areas as follows: Different Advanced Biofilm Detection and Imaging Techniques: The development of improved techniques for early detection and accurate imaging of biofilms will be crucial in diagnosing biofilm-related infections and monitoring treatment efficacy. This may involve the utilisation of advanced imaging modalities, such as high-resolution microscopy, molecular imaging, and non-invasive imaging techniques, to visualise biofilm structures and dynamics in real time. Targeted and Personalised Therapies: The future of biofilm treatment lies in the development of targeted and personalised therapies that consider the specific biofilm characteristics, microbial composition, and host factors. This may involve the use of genomics, proteomics, and metabolomics approaches to identify biomarkers associated with biofilm infections, allowing for tailored treatment strategies. Combination Therapies: Combining multiple treatment modalities, including traditional antimicrobials and biofilm-disrupting agents, holds promise for enhanced biofilm eradication. Synergistic combinations that target different stages of biofilm development and incorporate strategies to weaken the biofilm matrix and overcome antibiotic resistance are likely to be explored. Nanotechnology and Biomaterials: Continued research in nanotechnology and biomaterials will contribute to the development of novel strategies for biofilm control. Nanoparticles, nano-coatings, and biomaterial modifications can be designed to specifically target biofilms, disrupt their structure, and deliver antimicrobial agents in a controlled manner. Biofilm Engineering and Prevention Strategies: Engineering surfaces and materials with properties that inhibit biofilm formation or facilitate biofilm removal will be an important focus. This may involve the development of antimicrobial surface coatings, biofilm-resistant materials, and innovative approaches to prevent biofilm attachment and colonisation. Clinical Trials and Validation Studies: The efficacy and safety of novel treatment approaches targeting biofilms need to be rigorously evaluated through well-designed clinical trials and validation studies. These studies will provide essential evidence for the integration of alternative therapies into clinical practice and the development of standardised guidelines for biofilm management. To summarize, advancements can be made in detection techniques, personalised therapies, combination treatments, nanotechnology applications, biofilm engineering, and clinical validation studies. By pursuing these prospects, we can anticipate significant improvements in the management and control of biofilm-associated infections, ultimately improving patient outcomes and reducing the global burden of biofilm-related complications.

## 7. Conclusions

Microbial biofilms have significant implications across different sectors, ranging from the environment to healthcare. Traditional antibiotics are often ineffective against biofilms, and the development of antimicrobial resistance further complicates the available treatment options. Therefore, there is a need to adopt alternative strategies to control or eradicate biofilms. Understanding the applications and clinical consequences of biofilms is crucial for developing effective strategies to combat biofilm-related challenges. Alternative therapies that target biofilms show promising results in overcoming the limitations of traditional antimicrobial approaches. However, further research and clinical trials are necessary to fully evaluate their efficacy, safety, and potential integration into current treatment regimens. By advancing our knowledge of biofilm biology and exploring innovative therapeutic options, we can pave the way for improved management of biofilm-associated infections and enhance the overall efficacy of antimicrobial interventions.

## Figures and Tables

**Figure 1 microorganisms-11-01934-f001:**
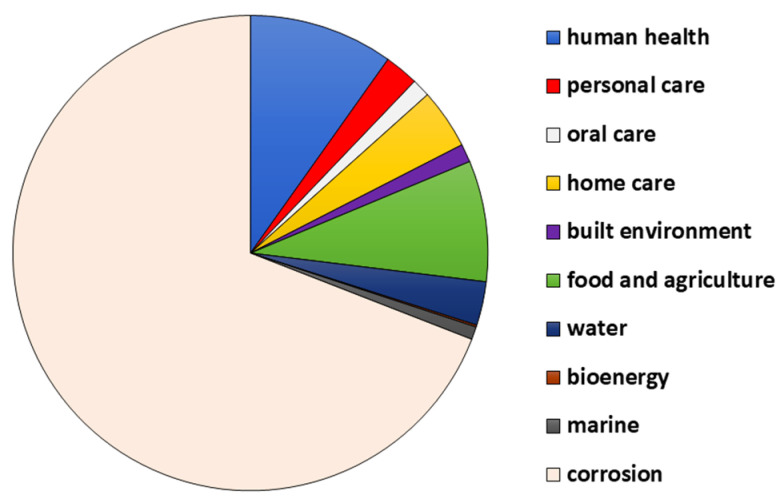
The distribution of cost (in percent) among various sectors.

**Figure 2 microorganisms-11-01934-f002:**
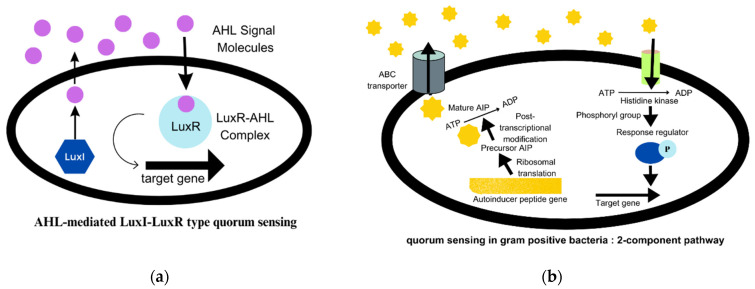
(**a**) AHL-mediated quorum sensing in Gram-negative bacteria. (**b**) Quorum signalling mechanism in Gram-positive bacteria.

**Figure 3 microorganisms-11-01934-f003:**
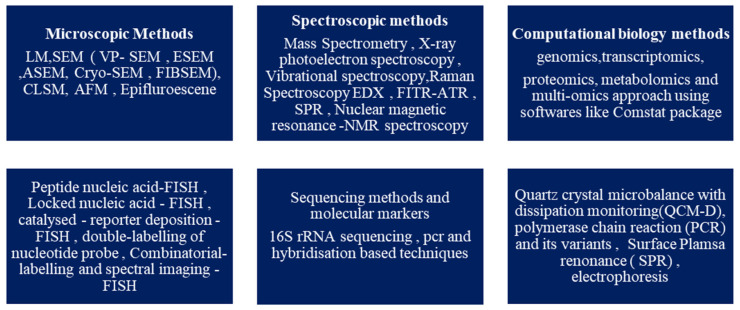
Characterisation techniques for biofilms.

**Figure 4 microorganisms-11-01934-f004:**
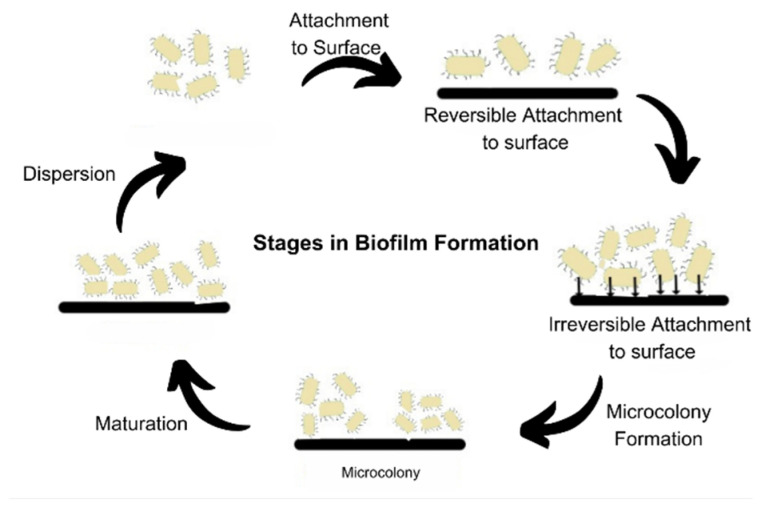
Stages of biofilm formation.

**Figure 5 microorganisms-11-01934-f005:**
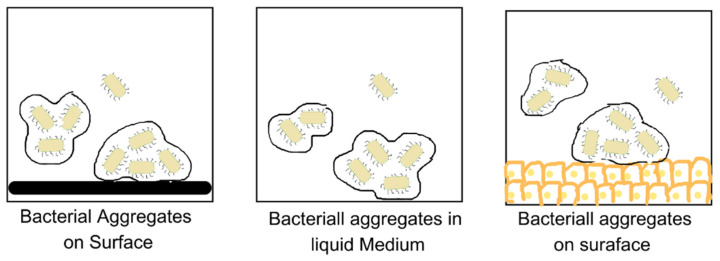
Bacterial aggregates on different kinds of surfaces.

**Figure 6 microorganisms-11-01934-f006:**
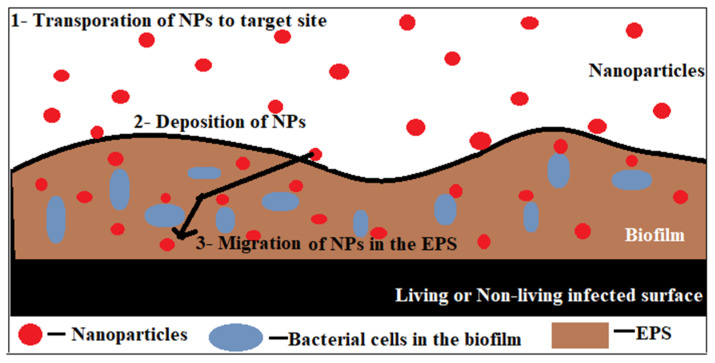
Targeting of biofilms by utilising a nanoparticle-mediated method.

**Figure 7 microorganisms-11-01934-f007:**
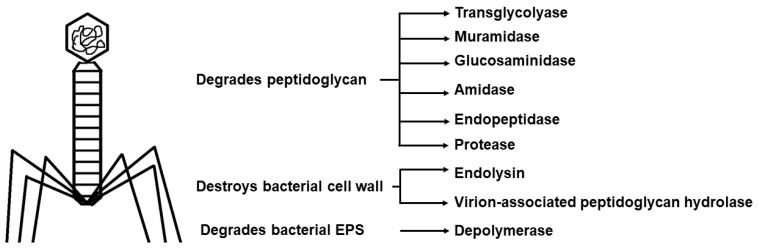
Enzymatic degradation by phages.

**Table 2 microorganisms-11-01934-t002:** Plant extracts tested for antibacterial properties.

Bacteria	Extract	MIC	Plant	Reference
*Staphylococcus aureus*	Methanol extract	1.25 mg/mL	*Allium sativum*	[136]
	Ethanol extract	2.5 mg/mL	*Allium sativum*	
	Hexane extract	5 mg/mL	*Cinnamomum verum*	[126]
	Dichloromethane extract	20 mg/mL	*Cinnamomum verum*	
	Ethanol extract	10 mg/mL	*Cinnamomum verum*	
	Clove oil	0.5 mg/mL	*Syzygium aromaticum*	
	Aqueous extract	0.5 mg/mL	*Solanum trilobatum*	[137]
*Bacillus cereus*	Methanol extract	0.156 mg/mL	*Allium sativum*	
	Ethanol extract	0.078 mg/mL	*Allium sativum*	
*Streptococcus pneumoniae*	Methanol extract	0.312 mg/mL	*Allium sativum*	
	Ethanol extract	0.312 mg/mL	*Allium sativum*	
*Pseudomonas aeruginosa*	Methanol extract	1.25 mg/mL	*Allium sativum*	
	Ethanol extract	0.625 mg/mL	*Allium sativum*	
	Essential oil	12–19 mg/mL	*Cinnamomum cassia*	[126]
	Ethanol extract	10 mg/mL	*Cinnamomum verum*	
	Dichloromethane extract	20 mg/mL	*Cinnamomum verum*	
	Hexane extract	10 mg/mL	*Cinnamomum verum*	
*Escherichia coli*	Methanol extract	0.625 mg/mL	*Allium sativum*	[136]
	Ethanol extract	0.156 mg/mL	*Allium sativum*	
	Essential oil	26–35 mg/mL	*Cinnamomum cassia*	[126]
	Clove oil	0.5 mg/mL	*Syzygium aromaticum*	
	Ethanol extract	0.39 mg/mL	*Syzygium aromaticum*	
	Essential oil	0.25 mg/mL	*Cuminum cyminum*	
	Ethanol extract	6.25 mg/mL (inhibition rate of 48.18% at MIC and eradication rate of 46.16% at 8 MIC)	Cinnamon	[138]
*Klebsiella pneumoniae*	Methanol extract	0.312 mg/mL	*Allium sativum*	[136]
	Ethanol extract	0.156 mg/mL	*Allium sativum*	
	Hexane extract	20 mg/mL	*Cinnamomum vernum*	[126]
	Dichloromethane extract	20 mg/mL	*Cinnamomum vernum*	
	Ethanol extract	20 mg/mL	*Cinnamomum vernum*	
	Essential oil	27–32 mg/mL	*Cinnamomum cassia*	
	Ethanol extract	0.78 mg/mL	*Syzygiumaromaticum*	
	Essential oil	0.8–3.5 mg/mL	*Cuminum cyminum*	
	Aqueous extract leaves	0.63 mg/mL	*Solanum trilobatum*	[137]
	Water, methanol, ethanol, and petroleum ether extract	160 µg/ml	*Adhatodavasica*	
*Enterobacter* spp.	Ethanol extract	0.78 mg/mL	*Syzygium aromaticum*	[126]
*Acinetobacter baumanii*	Ethanol extract	0.78 mg/mL	*Syzygium aromaticum*	
*Citrobacter* spp.	Ethanol extract	039 mg/mL	*Syzygium aromaticum5*	
*Enterococcus faecalis*	Essential oil	0.125 mg/mL	*Cuminum cyminum*	
	Ethanol extract	0.125 mg/mL	*Cuminium cyminum*	
	Methanol extract	9.63 mg/mL	*Piper nigrum*	
	Ethanol extract	100 mg/mL	*Salvia rosmarinus* (Rosemary)	
*Proteus mirabilis*	Methanol extract	9.63 mg/mL	*Piper nigrum*	
	Essential oil	30–39 mg/mL	*Cinnamomum cassia*	
	Ethanol extract	0.39 mg/mL	*Syzygium aromaticum*	
	Aqueous extract	32 µg/ml	*Piper betle*	
*Enterohemorrhagic Escherichia coli O157:H7*	Essential oil	3.12 µg/mL (Inhibition of biofilm was noticed at MIC/2 and MIC/4 concentrations)	*Thymus daenensis*	[139]
	Essential oil	6.25 µg/mL (Inhibition of biofilm was noticed at MIC/2 and MIC/4 concentrations)	*Satureja hortensis*	
*Vibrio parahaemolytics*	Ethanol extract	6.25 mg/mL (Inhibition rate of 75.46% at MIC and eradication rate of 93.26% at 32MIC)	Cinnamon	[138]
*Bacillus paramycoides*	Ethanolic extract	0.2514 µg/mL	*Zingiber officinale*	[140]

**Table 3 microorganisms-11-01934-t003:** Some selected nanoparticles useful in biofilm eradication.

Group	Type	Sub-Type	Characteristics	References
Organic	Liposomes	-	Advantages include target specificity, non-immunogenicity, low toxicity, biofilm matrix fusogenicity, adaptability for payloads, improvement of antimicrobial agent efficiency, and reduction of infection recurrence.	[147]
	Polymeric NPs	-	They show a strong antimicrobial nature, adaptable nature, and potential to penetrate biofilms of two species.	[148]
	Dendrimers	Cationic Dendrimers	Multivalency, well-organised structure, and solubility in water.	[149]
	Cyclodextrins	-	They can easily solubilise drugs and are poorly soluble in water, and can hence act as efficient modes of drug delivery.	[150]
	Solid–Lipid NPs	-	They provide low toxicity and more control over the release of drugs and a low cost of production.	
Inorganic	Metallic NPs	Gold	AuNPs and AgNPs disrupt bacterial membranes, interact with cytoplasmic contents, and induce oxidative stress by releasing ROS and disrupting the metabolic activities of the bacterial cell.	[151]
		Silver		
		Copper	It has an antimicrobial property and is often used in combination with other metallic nanoparticles, such as silver NPs.	[152]
		Silica	It is biocompatible, has a large surface area, and allows targeted drug delivery.	[153]
	Metal Oxides	Iron oxide	They are mainly used owing to their magnetic properties and high levels of biocompatibility.	[154]
		Copper oxide		
	Fullerene	-	Surfaces coated with fullerene have been seen to have less surface area infested with biofilm, and the formed biofilm has comparatively less biomasses.	[155]
	Quantum Dots		They have a small size, excellent biocompatibility, and cell permeability.	[156]

## Data Availability

No new data were generated during this study. Therefore, the concept of data availability does not apply to this article.

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
