# Peer review of "Microbial Biofilms: Applications, Clinical Consequences, and Alternative Therapies"

_microorganisms, 2023, doi:10.3390/microorganisms11081934_

Round 1

Reviewer 1 Report

This exhaustive reviwe focus on the description of the importance of biofilms in healthcare and provides information about new challenges and the potential use of phytotherapy to combat them.

My only suggestion is that they should divide subheadings according to device-related and tissue-related infections. And include and explain other infections that have not been already mentioned such as: VAP, CRBSI, PJI, contact lens infections, VVRC, other implants, ...

Author Response

Query 1:

My only suggestion is that they should divide subheadings according to device-related and tissue-related infections. And include and explain other infections that have not been already mentioned such as: VAP, CRBSI, PJI, contact lens infections, VVRC, other implants, ...

Response:

I extend my appreciation for the valuable suggestion provided. The text has been appropriately classified into tissue and implant-related infections in the manuscript. Furthermore, we have incorporated the suggested additional information on implant-related infections into the manuscript, as advised by the reviewer.

Reviewer 2 Report

After reading the manuscript titled "Microbial Biofilms: Challenges, Clinical Consequence, and Phytotherapy" I find it interesting, but still needing energy to polish.

1. Particularly noteworthy is the title, abstract and conclusions. I believe that it does not adequately reflect the content of the manuscript, because it does not focus only on phytotherapy, but also on anti-QS therapy (not only with plants), antimicrobial peptides and bacteriophages. I believe that the word "phytotherapy" should be replaced with "alternative therapies". If this is not possible, please delete sections not directly related to phytotherapy.

2. Cyanobacteria belong to bacteria, while in several places in the text (eg., line 38) they are presented as an independent class of microorganisms.

3. "Earth" should be capitalized. Please correct all places in the manuscript.

4. The text should be linguistically corrected. Some parts of the manuscript are difficult to understand, e.g., lines 40-43, 91, 331-335, 457-460 (but not limited only to these).

5. Some sentences are a repetition of the previous ones and should be deleted or modified, eg., lines 173-175 or 478-481.

6. Toxins and enzymes are virulence factors - logical mistake.

7. Lines 220-221, 271 : Some words, other than names of bacteria, should not be written using italics. Please correct.

8. Line 365: "The prebiotic biofilm" -> The probiotic biofilm

9. Line 449: "mainly sue" -> mainly due

10. In Table 1, it is difficult to tell where one line ends and the next begins. Please modify accordingly.

11. Table 2 should be extended to the whole width of the page (which is also in accordance with MDPI standards), because currently plant names are broken up into several lines and it is difficult to read.

12. Table 2: "Pseudomonas mirabilis" -> Proteus mirablis

13. Table 2: Klebsiella pneumoniae MTCC 3384 -> should be joined to the line with Klebsiella pneumoniae

14. Section "5.5. Quorum sensing inhibitors" should be joined to the section "5.3. Anti-virulence compounds from plants"

The text should be linguistically corrected. Some parts of the manuscript are difficult to understand, e.g., lines 40-43, 91, 331-335, 457-460 (but not limited only to these).

Author Response

Query 1:     

Particularly noteworthy is the title, abstract and conclusions. I believe that it does not adequately reflect the content of the manuscript, because it does not focus only on phytotherapy, but also on anti-QS therapy (not only with plants), antimicrobial peptides and bacteriophages. I believe that the word "phytotherapy" should be replaced with "alternative therapies". If this is not possible, please delete sections not directly related to phytotherapy.

Response:

We would like to express our gratitude for bringing the irregularities to our attention. We have made necessary modifications to the title, abstract, and conclusions of the manuscript to ensure that they accurately reflect the content.

Query 2:     

Cyanobacteria belong to bacteria, while in several places in the text (eg., line 38) they are presented as an independent class of microorganisms.

Response:

Considering the reviewers' observations, we have thoroughly reviewed the manuscript and consistently categorized the term "cyanobacteria" under the bacteria category throughout the manuscript.

Query 3:     

"Earth" should be capitalized. Please correct all places in the manuscript.

Response:

As per the reviewer’s suggestion, the first letter has been capitalized throughout the manuscript.

Query 4:     

The text should be linguistically corrected. Some parts of the manuscript are difficult to understand, e.g., lines 40-43, 91, 331-335, 457-460 (but not limited only to these).

Response:

Thank you for the observations provided. We have made efforts to simplify the language used in the mentioned sentences, as per your suggestion.

Query 5:     

Some sentences are a repetition of the previous ones and should be deleted or modified, eg., lines 173-175 or 478-481.

Response:

We have made necessary corrections to the lines in order to avoid the repetition of concepts/lines within the manuscript.

Query 6:     

Toxins and enzymes are virulence factors - logical mistake.

Response:

We appreciate the observations provided. As per the respected reviewer’s feedback, we have modified the terminology, replacing "Toxins and Enzymes" with "forms of virulence factors."

Query 7:     

Lines 220-221, 271 : Some words, other than names of bacteria, should not be written using italics. Please correct.

Response:

We have made the necessary revisions as per the reviewer’s suggestion to change all words, except for the names of plants and bacteria, to normal font.

Query 8: 

Line 365: "The prebiotic biofilm" -> The probiotic biofilm

Response:

The spelling has been corrected.

Query 9:                                                                                    

Line 449: "mainly sue" -> mainly due

Response:

The correction has been made.

Query 10:     

In Table 1, it is difficult to tell where one line ends and the next begins. Please modify accordingly.

Response:

Thank you for providing your feedback. We have made the required formatting adjustments to enhance the clarity of the table.

Query 11:     

Table 2 should be extended to the whole width of the page (which is also in accordance with MDPI standards), because currently plant names are broken up into several lines and it is difficult to read.

Response:

In accordance with the suggestion of the reviewer, the tables have been expanded to occupy the entire width of the page.

Query 12:     

Table 2: "Pseudomonas mirabilis" -> Proteus mirabilis

Response:

Thank you for the accurate observation. We have corrected the mistake accordingly.

Query13:
Table 2: Klebsiella pneumoniae MTCC 3384 -> should be joined to the line with Klebsiella pneumoniae

Response:                                                       

Thank you for your valuable feedback. The modifications have been done.

Query14:
Section "5.5. Quorum sensing inhibitors" should be joined to the section "5.3. Anti-virulence compounds from plants"

Response:                      

As per the suggestion, the two sub-sections have been merged.

Reviewer 3 Report

1. The authors state in the title, that the article concerns the phytotherapy of bacterial biofilms, yet several subsections describe such methods as phage therapy and quorum sensing inhibitors. It is not clear, whether the purpose of the review was to describe other methods of biofilm treatment, not limited to phytotherapy. If so, the authors should change the title, suggesting the review being not limited to phytotherapy-based methods, and add several more well-known methods, such as the use of nanoparticles etc.
2. The comparison of the phytoextracts’ efficiency to the standard antibiotic treatment should be made.
3. While talking about phytoextracts, the authors should state clearly, whether the biofilms, used in the cited works, were multidrug-resistant or not.
4. The quality of the figures needs to be improved.

Author Response

Query 1:     

The authors state in the title, that the article concerns the phytotherapy of bacterial biofilms, yet several subsections describe such methods as phage therapy and quorum sensing inhibitors. It is not clear, whether the purpose of the review was to describe other methods of biofilm treatment, not limited to phytotherapy. If so, the authors should change the title, suggesting the review being not limited to phytotherapy-based methods, and add several more well-known methods, such as the use of nanoparticles etc.

Response:

Thank you for your valuable feedback. In response to your input, we have made revisions to the title, abstract, and conclusion sections of the manuscript. These sections have been updated to encompass additional well-established therapeutic approaches, particularly on plant-derived compounds. Furthermore, in accordance with the suggestions provided by the esteemed reviewer, we have incorporated nanotechnology-based techniques into our discussion.

Query 2:     

The comparison of the phytoextracts’ efficiency to the standard antibiotic treatment should be made.

Response:

In accordance with the suggestion of the reviewer, we have conducted a comparison of the efficiency of phytoextracts with standard antibiotics in the section dedicated to phytoextracts.

Query 3:     

While talking about phytoextracts, the authors should state clearly, whether the biofilms, used in the cited works, were multidrug-resistant or not.

Response:

We appreciate the reviewer's valuable comment. In the section discussing phytoextracts, we have included a clear statement regarding the multidrug resistance status of the biofilms used in the cited works. This clarification will enhance the clarity and relevance of our discussion. Thank you for bringing this important point to our attention.

Query4:
The quality of the figures needs to be improved.

Response:                                                       

Thank you for your valuable suggestion. We have taken it into consideration and made the necessary improvements to enhance the quality of the figures. As a result, all the figures in the manuscript are now clearer and more visually appealing.

Round 2

Reviewer 2 Report

The authors put a lot of effort into improving the manuscript and making appropriate corrections, which resulted in a significant increase in its quality.

During the final corrections (author's proofreading), I ask you to take into account two minor modifications:

- line 168: "metastatic seeding" -> "seeding dispersal" (metastasis is reserved for tumor cells)

- lines 647-663: in this new section, please use italics in all the names of bacteria or plants

The language side of the manuscript has improved.

Reviewer 3 Report

The quality of the manuscript has improved significantly. 

Still, there are some issues with typos and incorrect use of the uppercase (e.g., line 698 - "Biofilm" and "Nanoparticles"; line 710 - "Metal Oxides" etc., which should be addressed to during the proofreading of the manuscript.